# Clinical bleeding patterns and management techniques of abnormal uterine bleeding at a teaching and referral hospital in Western Kenya

Godfrey Shichenje Mutakha[ID]*, Emily Mwaliko, Philip Kirwa

Department of Reproductive Health, School of Medicine – Moi University, Eldoret, Kenya

* godfreymutakha@gmail.com

## Abstract

Abnormal uterine bleeding (AUB) affects 30% of reproductive age women globally. However, there are limited local studies evaluating the management of these women. The diagnostic guideline using structural and functional causes of AUB adopts the PALM-COEIN criteria, namely: **P**olyp; **A**denomyosis; **L**eiomyoma; **M**alignancy and Hyperplasia; **C**oagulopathy; **O**vulatory dysfunction; **E**ndometrial; **I**atrogenic; and **N**ot yet classified. This study aimed to determine the clinical bleeding patterns, adherence to PALM-COEIN diagnosis guidelines and management of AUB among women in their reproductive age. This was a cross-sectional study among 108 women with AUB aged 18–45 years was conducted at the gynaecology department of Moi Teaching and Referral Hospital in Western Kenya between April 2018 and April 2019. Their sociodemographic and clinical characteristics were collected using interviewer administered structured questionnaire and chart reviews. Adherence to diagnosis recommendations was assessed using PALM COEIN classification. Descriptive and inferential data analysis was conducted at 95% confidence interval. The median age was 30 (IQR: 22, 41) years with prolonged bleeding as the most predominant pattern at 41.7%. Bleeding patterns were significantly associated with age (p = 0.04). Only 16.7% were diagnosed as per the PALM-COEIN criteria with PALM and COEIN accounting for 60% and 40% respectively. Leiomyoma (44.5%) was the common cause of AUB. Laboratory evaluation included: pregnancy tests, full haemogram, hormonal profile and biopsy. Most (79.6%) of the women had abdominopelvic ultrasound done. Medical management was provided for 78.7% of women. Prolonged bleeding was the most common pattern with medical management preferred. There is need for in-hospital algorithms to ensure adherence to PALM-COEIN guidelines.

## Introduction

Abnormal uterine bleeding (AUB) among women has a global prevalence of between 3–30% [1] accounting for about one third of outpatient gynaecology visits. This condition affects the

**Data Availability Statement:** All relevant data are within the manuscript and Supporting information files.

**Funding:** The author(s) received no specific funding for this work.

**Competing interests:** The authors have declared that no competing interests exist.

quality of life for women with socioeconomic and psychological consequences [1]. Its occurrence is dependent on temporal and quantitative regulation of reproductive hormones (hypothalamic-pituitary-ovarian axis). The menstrual flow mediated by prostaglandins occurs following degeneration of the corpus luteum. Currently, the use of PALM-COEIN (Polyps; Adenomyosis; Leiomyoma; Malignancy and Hyperplasia; Coagulopathy; Ovulatory dysfunction; Endometrial; Iatrogenic; and Not yet classified) classification reduces the general inconsistency in the description of AUB in clinical and research settings.

Although the PALM-COEIN classification guideline is the gold standard for AUB diagnosis; it is not commonly used in many clinical settings leading to a lack of standardization in the diagnosis of women with AUB. There is limited documented studies on the bleeding patterns and management guidelines for women with AUB in Kenya. There is need to determine the proportions of women presenting with AUB at MTRH to plan for various management options. Knowledge of clinical presentation patterns of AUB will be used in putting in place control strategies. Evaluation of management strategies and their conformity to guidelines will create new knowledge on what aspects to be modified.

Clinical bleeding patterns are determined by the heaviness, duration of flow, regularity, and frequency. The causes of AUB can either be structural (PALM) or non-structural (COEIN) [2]. Management on the other hand includes the diagnostic techniques and therapeutic interventions offered to the affected women [3]. Diagnosis could be through laboratory, radiological and other imaging techniques [4]. Clinical management for women with AUB is either medical or surgical. Medical management is the first line therapeutic option once malignancy and pelvic pathology have been ruled out [5]. On the other hand, Surgical Management includes both minimal invasive techniques such as endometrial ablation for heavy menstrual bleeding [6] and invasive techniques such as hysterectomy and myomectomy. Hysterectomy is the definitive solution with high rates of patient satisfaction [7]. This study aimed to determine the clinical bleeding patterns and management of AUB among women in their reproductive age. Specifically, it determined level of adherence to PALM-COEIN diagnostic guidelines for uterine bleeding, diagnostic tests ordered, medical and surgical management offered for women presenting with abnormal uterine bleeding.

## Materials and methods

This was a cross-sectional study among women presenting with abnormal uterine bleeding at Moi Teaching and Referral Hospital in Western Kenya from April 2018 to April 2019. The facility is the second largest tertiary hospital in Kenya catering for patients in Western and North Rift Valley regions of the country. A census was conducted among all the reproductive age (18–45 years) women seeking care for abnormal uterine bleeding. This was defined as an episode of bleeding in a woman of reproductive age, who is not pregnant and is of sufficient quantity to require immediate intervention to prevent further blood loss. If the bleeding prolonged over a six-month period, it was referred to as chronic abnormal uterine bleeding. Patient's history and clinical information from medical records were collected. These included: the diagnostic tests done, final diagnosis made, and treatment given. Data analysis was conducted using statistical package for social sciences (SPSS) version 22 software. Descriptive statistics were used to summarize categorical variables such as level of education and marital status. Continuous variables such as age, duration of bleeding among others were summarized using mean and the corresponding standard deviation if the Gaussian assumptions hold. Otherwise they were summarized using median and the corresponding inter quartile range (IQR). Gaussian assumptions were assessed using histograms and the normal probability plots. Inferential statistics techniques using Pearson chi-square test was conducted to determine the level

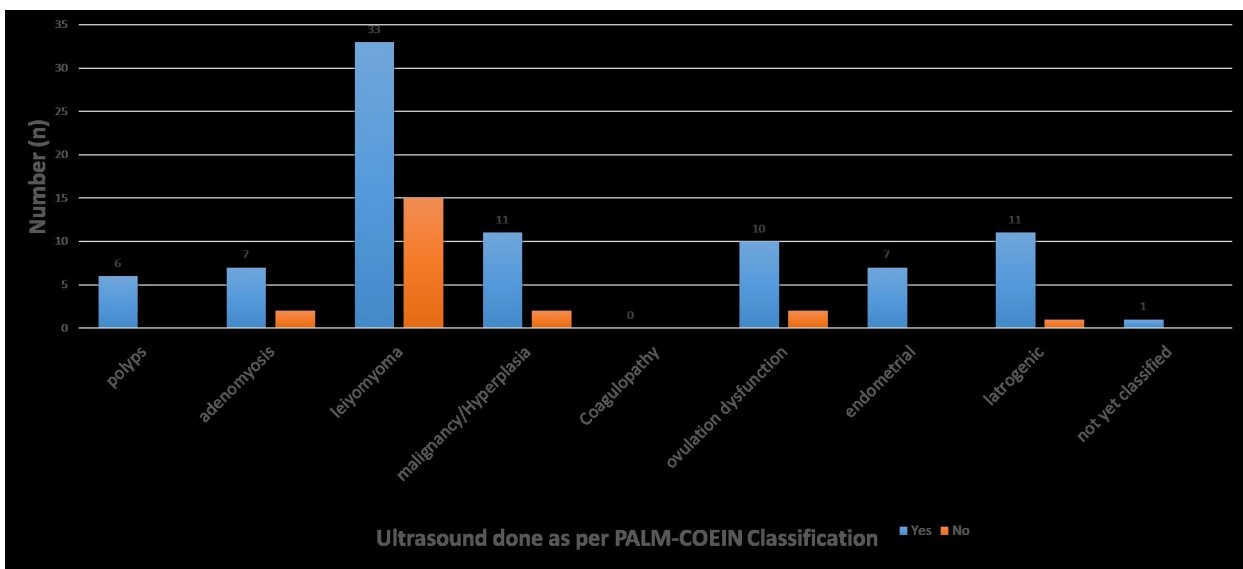

**Fig 1. Ultrasound done as per the PALM-COEIN criteria.**

of statistical significance between predictor and outcome variables. The study's ethical approval was obtained from the Institutional Research and Ethics Committee of Moi University and Moi Teaching and Referral Hospital.

## Results

The study enrolled a total of 108 women with AUB as per the flow chart on Fig 1. The mean age of the study participants was 31.46 years (SD ± 11.17). Most 64.8% (n = 70) of them had attained secondary level of education with 24.1% (26) having attended university/college level of education. Only 10.2% (n = 11) were formally employed while more than half (57.4%; n = 62) of all the participants living outside Eldoret.

### Abnormal uterine bleeding patterns

Prolonged bleeding was the most common (41.7%) bleeding pattern followed by heavy (35.2%), frequent (20.4%), intermenstrual (18.5%) and infrequent (8.3%) bleeding as (Table 1).

### Management of women with AUB

Management of women with AUB was categorized in terms of diagnosis and cost of management.

**Table 1. Abnormal uterine bleeding patterns seen at MTRH.**

| Bleeding pattern | Frequency | Percentage |
|---|---|---|
| Prolonged bleeding | 45 | 41.7 |
| Heavy bleeding | 38 | 35.2 |
| Frequent bleeding | 22 | 20.4 |
| Intermenstrual bleeding | 20 | 18.5 |
| Infrequent bleeding | 9 | 8.3 |

**Table 2. Diagnosis as per the PALM-COEIN criteria.**

| Diagnosis | P | A | L | M | C | O | E | I | N | Totals |
|-----------|---|---|---|---|---|---|---|---|---|--------|
| Yes | 1 | 1 | 13 | 1 | 0 | 2 | 0 | 0 | 0 | 18 (16.7%) |
| No | 5 | 8 | 35 | 12 | 0 | 10 | 7 | 12 | 1 | 90 (83.3%) |
| Total | 6 | 9 | 48 | 13 | 0 | 12 | 7 | 12 | 1 | 108 (100%) |

(p-value = 0.364).

Out of the 108 participants enrolled, only 16.7% (n = 18) were diagnosed as per the PALM-COEIN criteria. However, there was no statistically significant difference (p-value = 0.364) in abnormal uterine bleeding between the groups that were diagnosed as per the PALM-COEIN criteria versus those that were not (Table 2).

All the study participants had a pregnancy test done. This was followed by a complete blood count (CBC) test among nearly half (45.3%; n = 49), coagulation profile (25.9%; n = 28) and Thyroid Stimulating Hormone (TSH) at 12% (n = 13).

Additional hormonal profile laboratory tests were conducted based on menstrual cycle pattern (regular versus irregular). Overall, follicle stimulating hormone (FSH) was the most commonly (13%) ordered among all the study participants with 13.2% of the women with regular menstrual cycles being subjected to it. This was followed by a TSH test that was commonly ordered for women with irregular (15%) compared to those with regular (10.3%) cycles. Prolactin hormone test was the least frequently ordered. Estradiol and progesterone hormone tests were not ordered at all. There were no statistically significant relationships reported between menstrual cycle patterns (Table 3). Biopsy sampling was categorized by the participants age and whether it was done. Among women aged 35 years or less (n = 72), none of them had a biopsy specimen collected. On the other hand, those aged more than 35 years, only 8.3% (n = 3) had biopsy sampling done.

As per the PALM-COEIN classification, all women presenting with polyps, endometrial causes and non-classified abnormal uterine bleeding had a pelvic ultrasound done. Leiomyoma was the most commonly presenting cause of endometrial bleeding, of which more than two thirds (68.8%; n = 33) of the women presenting with it had a pelvic ultrasound ordered. Higher proportions of pelvic ultrasound requests were reported among those with adenomyosis (77.8%; n = 7), malignancy/hyperplasia (84.6%; n = 11), ovulation dysfunction (83.3%; n = 10) and iatrogenic (91.7%; n = 11) causes of abnormal uterine bleeding (Fig 1). Even though 8.3% (n = 9) of the participants presented with adenomyosis; none of them got an MRI scan done.

Majority of the women presenting with AUB were treated medically. Analgesics were given to nearly all (92.6%; n = 100) the study participants while more than three quarters (78.7%; n = 85) received antibiotics. Tranexamic acid (TXM) was prescribed to more than half (61.1%; n = 66) of the study participants, followed by haematinics (39.8%; n = 43) while blood transfusion and combined oral contraceptives (COCs) were given to 6.5% (n = 7) and 5.5% (n = 6)

**Table 3. Laboratory testing of hormonal profiles as per the menstrual cycles.**

| Cycle | TSH | Estradiol | Prolactin | Progesterone | FSH | LH |
|-------|-----|-----------|-----------|--------------|-----|-----|
| Regular (n = 68) | 7 (10.3%) | 0 | 1 (1.5%) | 0 | 9 (13.2%) | 0 |
| Irregular (n = 40) | 6 (15%) | 0 | 2 (5%) | 0 | 5 (12.5%) | 0 |
| Total (N = 108) | 13 (12%) | 0 | 3 (2.8%) | 0 | 14 (13%) | 0 |
| p-value | 0.468 | - | 0.281 | - | 0.913 | - |

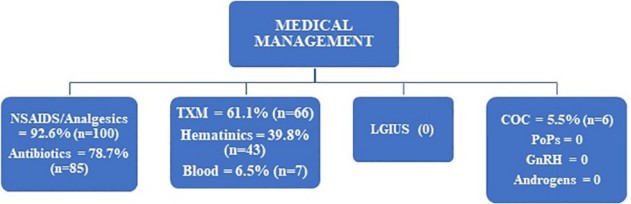

**Fig 2. Medical management chart for women with AUB.**

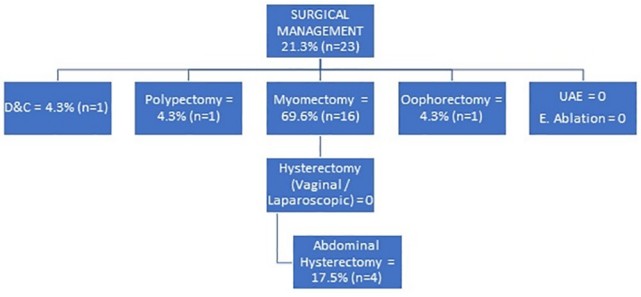

**Fig 3. Surgical management of women with AUB.**

respectively. No participant was given levonorgestrel intrauterine system (LGIUS), Progesterone only Pills (POP), androgens and gonadotropin releasing hormone (GnRH) agonists as (Fig 2).

Surgical management was offered to 21.3% (n = 23) of all the study participants, majority (69.6%; n = 16) of whom got myomectomy done. This was followed by abdominal hysterectomy at 17.5% (n = 4). The less frequent procedures were dilatation and curettage (D&C), polypectomy and oophorectomy at 4.3% (n = 1) each. No participant got uterine artery embolization (UAE), endometrial Ablation and vaginal or laparoscopic hysterectomy (Fig 3).

## Discussion

The study was undertaken among women of reproductive age to determine the causes of abnormal uterine bleeding based on PALM-COEIN classification and the management options adopted. Nearly two-thirds of the study participants had a minimum of secondary education; however only one-tenth of those enrolled were engaged in formal employment. This could be attributed to the low unemployment rates in Kenya especially among women of reproductive age who mainly engage in informal businesses or take care of their families as housewives. This was confirmed by a national survey of Kenyan youth (aged 18–35 years) where 62% of women were unemployed compared to an average unemployment rate of 15% among youths with a minimum of secondary education [8].

This study reported that prolonged bleeding was the most common bleeding pattern seen among nearly half (41.7%; n = 45) of all the women enrolled into the study. This finding was higher than an Indian study [9] with one-third (33.6%) of all the study participants presenting with prolonged bleeding. The Indian study adopted a retrospective study design over a six-month period among 250 patients aged between 25 to 65 years. This variance in study design, period and target population could explain the variance in proportions of study participants.

Prolonged bleeding was defined as more than eight (8) days of bleeding. It could have been easier to count more than 8 days than quantify other forms of AUB such as heavy bleeding.

Heavy Bleeding was the second most common AUB pattern among more than one-third (35.2%; n = 38) of all enrolled study participants. This finding is matched a Brazilian study that reported a proportion of women with heavy bleeding at 35.3% [10]. Both studies adopted cross-sectional study designs among women aged 18–45 years and this could explain the similarity in proportions.

Frequent bleeding was the third most frequent AUB pattern among nearly one-fifth (18%; n = 22) of all the study participants. This proportion is similar to the findings in another Indian study where 17% of all the women enrolled complained of frequent bleeding [11]. This similarity could be attributed to the fact that both studies targeted women in their reproductive years and were both conducted in teaching hospitals.

Less than one-fifth (16.7%; n = 18) of all the study participants were diagnosed as per the PALM-COEIN criteria. Preoperative classification of AUB as per PALM-COEIN classification system was correctly done among 130 (65.0%) of the 200 women in a study in a low resource setting [12]. However, the overall proportion of participants who met the PALM and COEIN criteria accounted for 60% (n = 64) and 40% (n = 44) respectively. This matched an Indian study [13] where 60.4% and 39.6% were diagnosed as per the PALM and COEIN criteria respectively. This similarity could be attributed to the fact that both studies were done in a tertiary teaching hospital among non-pregnant women over a period of one year. In a retrospective analysis of women who underwent a planned abdominal hysterectomy for AUB, leiomyoma was most common cause at 44.2% [14]. This finding is comparable to the current study's finding of nearly half (44.4%) of all study participants presenting with leiomyoma as the most common cause of abnormal uterine bleeding.

Malignancy and hyperplasia were the second most common abnormalities with a proportion of 12% (n = 13). There were equal proportions iatrogenic and ovulation dysfunction among the women sampled at 11.1% (n = 12). This matched an Indian study [11] where 13.6% of the study participants presented with malignancy and hyperplasia.

The most frequent laboratory tests conducted among the enrolled women with abnormal uterine bleeding were pregnancy tests (100%), complete blood counts (45.3%) and coagulation profile (25.9%), follicle stimulating hormone (13%) and thyroid stimulating hormone (12%). There were low proportions of and prolactin hormone (2.8%) tests. However, no participant got laboratory requests for oestradiol, progesterone and luteinising hormone. Pregnancy test is mandatory in the evaluation of women with AUB to rule out pregnancy related bleeding; while complete blood count was assayed to evaluate any associated infections, anaemia and low platelet count. Hormonal profiles were ordered to rule out abnormal uterine bleeding related to imbalance and to inform treatment options. Only 3 (2.7%) participants had a biopsy sample taken. This was done only among women aged below 35 years.

Abdominopelvic ultrasound was done for the majority (79.6%; n = 86) of the study participants to rule out structural causes of uterine bleeding. Those (20%) who did not have the ultrasound scans done were due to lack of funds or clinicians not requesting. This finding compares to a Pakistani study [15] in which all the study participants had an abdominopelvic ultrasound done. However, the findings contrast those in the Netherlands where only 10% of the women presenting with AUB had an abdominopelvic ultrasound conducted [16].

Medical and Surgical management was offered to 78.7% (n = 85) and 21.3% (n = 23) respectively. Analgesics were prescribed to nearly all (92.6%) of the study participants followed by antibiotics (78.7%) and antifibrinolytics (61.1%). Abdominal pain coupled with per vaginal bleeding are the common clinical presentation of abnormal uterine bleeding. Because majority of the women presented with intense abdominal pain, they were prescribed for analgesics at

the initial stage of treatment alongside antibiotics because of pyrexia and other infection symptoms. This finding contrasted a Dutch study [16] where nearly two thirds (62%) of the study participants did not receive any medication, while analgesics were prescribed among 5% of the study participants who presented with excessive bleeding. This could be attributed to the fact that majority of the women were initially seen by General Practitioners and only a few referred to gynecologists. A systematic review [17] by the Vanderbilt Evidenced based practice center of six studies on non-steroidal anti-inflammatory drugs (NSAIDS) among women with AUB; reported that NSAIDS reduced bleeding in the participants enrolled in all the six studies. The study further reported that tranexamic acid (TXA) was more effective in reducing bleeding than NSAIDS in three out of four studies reviewed.

Hematinics used at 43(39.8%) while blood transfusion done in 7(6.5%) although more than 56% needed them. Combined oral contraceptives were prescribed among 5.5% of the study participants enrolled in this study; because of their hormonal nature, they are effective in controlling abnormal uterine bleeding by regulating the menstrual cycle and reducing the flow volume. This was lower than the 35% of women who received hormonal treatment in the Netherlands [16]. A treatment algorithm for AUB [18] recommended that abnormal bleeding persists after three months, a higher dose of oral contraceptives can be used.

Myomectomy was done in more than two-thirds of those who received surgical management due to the fact that fibroids was the most common structural cause of bleeding; its fertility sparing benefit and popularity among the study participants who were women of reproductive age. This finding corroborates with that reported in a systematic review [19] which reported that among women with myomas at risk of abnormal uterine bleeding; myomectomy is a gold-standard treatment option for those women who wish to preserve their fertility. This is coupled by its safety, minimal invasiveness and greater patient satisfaction [19,20] as corroborated in the current study's findings.

## Conclusions and recommendations

We report that prolonged bleeding was the most common bleeding pattern followed by heavy bleeding among women of reproductive age seeking care at a teaching hospital in Western Kenya. There were low proportions (16.7%) of adherence to PALM-COEIN classification as a diagnostic guideline for women with abnormal uterine bleeding. These findings create a need for clinical evaluation of abnormal uterine bleeding among women who present with prolonged bleeding. More training and development of in-hospital clinical care algorithms should be done to ensure adherence to PALM-COEIN guidelines in the diagnosis of abnormal uterine bleeding. Medical management should be opted for as the first line of management for women presenting with abnormal uterine bleeding.

### What is already known on this topic

- Abnormal uterine bleeding affects women of reproductive age.

- The recommended diagnostic guideline is using the PALM-COEIN classification.

- There is limited documented studies on the bleeding patterns and management guidelines for women with AUB in Kenya.

- Management of AUB has been reported to be costly by studies done in other settings, further complicating patient management and outcomes.

## What this study adds

- Prolonged and heavy bleeding are the most common abnormal uterine bleeding patterns in Western Kenya.

- This study shows that the PALM-COEIN diagnostic criteria is not commonly used in Western Kenya.

- The study further compares PALM-COEIN diagnostic criteria with the management offered for women with AUB in Western Kenya.

## Supporting information

**S1 Dataset.**
(XLSX)

**S1 Appendix. Questionnaire.**
(DOCX)

## Acknowledgments

Moi Teaching and Referral Hospital management and health records department alongside Moi University Department of Reproductive Health.

## Author Contributions

**Conceptualization:** Godfrey Shichenje Mutakha, Emily Mwaliko, Philip Kirwa.

**Data curation:** Godfrey Shichenje Mutakha, Emily Mwaliko, Philip Kirwa.

**Formal analysis:** Godfrey Shichenje Mutakha.

**Supervision:** Emily Mwaliko, Philip Kirwa.

**Visualization:** Godfrey Shichenje Mutakha.

**Writing – original draft:** Godfrey Shichenje Mutakha, Emily Mwaliko, Philip Kirwa.

**Writing – review & editing:** Godfrey Shichenje Mutakha, Emily Mwaliko, Philip Kirwa.

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
