## [Decision Letter · Decision Letter 0]

27 Aug 2020

PONE-D-20-16619

CLINICAL BLEEDING PATTERNS AND MANAGEMENT TECHNIQUES OF ABNORMAL UTERINE BLEEDING IN WESTERN KENYA.

PLOS ONE

Dear Dr. Mutakha,

Thank you for submitting your manuscript to PLOS ONE. After careful consideration, we feel that it has merit but does not fully meet PLOS ONE’s publication criteria as it currently stands. Therefore, we invite you to submit a revised version of the manuscript that addresses the points raised during the review process.

We look forward to receiving your revised manuscript.

Kind regards,

Onikepe Oluwadamilola Owolabi

Academic Editor

PLOS ONE

2. Please include additional information regarding the survey or questionnaire used in the study and ensure that you have provided sufficient details that others could replicate the analyses.

For instance, if you developed a questionnaire as part of this study and it is not under a copyright more restrictive than CC-BY, please include a copy, in both the original language and English, as Supporting Information. 

If the original language is written in non-Latin characters, for example Amharic, Chinese, or Korean, please use a file format that ensures these characters are visible.

Reviewers' comments:

Reviewer's Responses to Questions

**Comments to the Author**

1. Is the manuscript technically sound, and do the data support the conclusions?

Reviewer #1: Partly

2. Has the statistical analysis been performed appropriately and rigorously? 

Reviewer #1: N/A

3. Have the authors made all data underlying the findings in their manuscript fully available?

Reviewer #1: Yes

4. Is the manuscript presented in an intelligible fashion and written in standard English?

Reviewer #1: Yes

5. Review Comments to the Author

Reviewer #1: Discussing AUB in Kenya is a matter of interest, however, conclusions based on 108 reported cases give a weak and no representative information for that.

In INTRODUTION, authors should sate the interest of the present study, but that information is not clear. For example: the difference between patterns of bleeding and causes of AUB seems confusing.

OBJECTIVES are not well defined. It’s not clear what the authors really intent to investigate

In MATERIALS and METHODS is missing important information, like the adopted definitions of AUB patterns and software used for statistical analysis of the data.

In RESULTS, characteristics of the studied population raise some doubts (64,8% had secondary level of education, but only 10,2% have an employment – this should be discussed). In AUB medical management is stated that 78,7% of the patients received antibiotics: this should be discussed.

In DISCUSSION, results are compared with those reported with other studies, but there is no explanation about the results obtained by the authors (why analgesics were given to nearly all patients? why myomectomy was performed in 69,9% of the patients? and so on…)

In CONCLUSIONS, the only one conclusion of this manuscript is that PALM-COEIN Guidelines are rarely used in Western Kenya, as in many parts of the world.

Thinking about knowledge and clinical practice applicability this study doesn’t bring anything new, however the data obtained by the authors is interesting and liable for constructing an informative and didactic paper.

6. PLOS authors have the option to publish the peer review history of their article (what does this mean?). If published, this will include your full peer review and any attached files.

Reviewer #1: No

---

## [Author Response · Author response to Decision Letter 0]

29 Sep 2020

Comment 1: Discussing AUB in Kenya is a matter of interest, however, conclusions based on 108 reported cases give a weak and no representative information for that.

This study was conducted in a teaching and referral hospital in Western Kenya. The tertiary healthcare facility offers clinical management for abnormal uterine bleeding alongside other gynaecological interventions to women from the greater Western Kenya. This makes the sample representative of the population, however not the whole country. To clarify this further, we have modified the study title to read “Clinical bleeding patterns and management techniques of abnormal uterine bleeding at a teaching and referral hospital in Western Kenya.”

Comment 2: In INTRODUTION, authors should state the interest of the present study, but that information is not clear. For example: the difference between patterns of bleeding and causes of AUB seems confusing.

This has been modified appropriately. The interest of the study has been highlighted in the second paragraph of the introduction. 

The difference between patterns and causes of bleeding have been differentiated as:

“Clinical bleeding patterns are determined by the heaviness, duration of flow, regularity, and frequency. 

The causes of AUB can either be structural (PALM) or non-structural (COEIN). Management on the other hand includes the diagnostic techniques and therapeutic interventions offered to the affected women.”

Comment 3: OBJECTIVES are not well defined. It is not clear what the authors really intent to investigate.

This has been modified as “This study aimed to determine the clinical bleeding patterns and management of AUB among women in their reproductive age. Specifically, it determined level of adherence to PALM-COEIN diagnostic guidelines for uterine bleeding, diagnostic tests ordered, medical and surgical management offered for women presenting with abnormal uterine bleeding.”

Comment 4: In MATERIALS and METHODS is missing important information, like the adopted definitions of AUB patterns and software used for statistical analysis of the data.

Abnormal uterine bleeding has been defined as: “an episode of bleeding in a woman of reproductive age, who is not pregnant and is of sufficient quantity to require immediate intervention to prevent further blood loss.” Bleeding Patterns were either heavy, light, frequent, infrequent, regular, irregular, prolonged or short.

Data analysis was conducted using statistical package for social sciences (SPSS) version 22 software.

Comment 5: In RESULTS, characteristics of the studied population raise some doubts (64.8% had secondary level of education, but only 10.2% have an employment – this should be discussed). In AUB medical management is stated that 78.7% of the patients received antibiotics: this should be discussed.

These have been discussed as follows:

“Nearly two-thirds of the study participants had a minimum of secondary education; however only one-tenth of those enrolled were engaged in formal employment. This could be attributed to the low unemployment rates in Kenya especially among women of reproductive age who mainly engage in informal businesses or take care of their families as housewives. This was confirmed by a national survey of Kenyan youth (aged 18-35 years) where 62% of women were unemployed compared to an average unemployment rate of 15% among youths with a minimum of secondary education.”

“Because majority of the women presented with intense abdominal pain, they were prescribed for analgesics at the initial stage of treatment alongside antibiotics because of pyrexia and other infection symptoms.”

 

Comment 6: In DISCUSSION, results are compared with those reported with other studies, but there is no explanation about the results obtained by the authors (why analgesics were given to nearly all patients? why myomectomy was performed in 69,9% of the patients? and so on…)

This has been modified appropriately as follows:

“Abdominal pain coupled with per vaginal bleeding are the common clinical presentation of abnormal uterine bleeding. Because majority of the women presented with intense abdominal pain, they were prescribed for analgesics at the initial stage of treatment alongside antibiotics because of pyrexia and other infection symptoms.”

“Myomectomy was done in more than two-thirds of those who received surgical management due to the fact that fibroids was the most common structural cause of bleeding; its fertility sparing benefit and popularity among the study participants who were women of reproductive age.”

Comment 7: In CONCLUSIONS, the only one conclusion of this manuscript is that PALM-COEIN Guidelines are rarely used in Western Kenya, as in many parts of the world.

The conclusion section has been expanded to include a second conclusion. Furthermore, the study recommendations have also expanded and modified appropriately. 

Other comments:

i. The manuscript structure has been updated in line with all PLoS requirements.

ii. Dataset has been uploaded for review.

iii. The study’s questionnaire has been appended as additional information.

---

## [Decision Letter · Decision Letter 1]

17 Nov 2020

Clinical bleeding patterns and management techniques of abnormal uterine bleeding at a teaching and referral hospital in Western Kenya.

PONE-D-20-16619R1

Dear Dr. Mutakha,

We’re pleased to inform you that your manuscript has been judged scientifically suitable for publication and will be formally accepted for publication once it meets all outstanding technical requirements.

Kind regards,

Onikepe Oluwadamilola Owolabi

Academic Editor

PLOS ONE

Additional Editor Comments (optional):

Reviewers' comments:

Reviewer's Responses to Questions

**Comments to the Author**

1. If the authors have adequately addressed your comments raised in a previous round of review and you feel that this manuscript is now acceptable for publication, you may indicate that here to bypass the “Comments to the Author” section, enter your conflict of interest statement in the “Confidential to Editor” section, and submit your "Accept" recommendation.

Reviewer #1: All comments have been addressed

2. Is the manuscript technically sound, and do the data support the conclusions?

Reviewer #1: Yes

3. Has the statistical analysis been performed appropriately and rigorously? 

Reviewer #1: Yes

4. Have the authors made all data underlying the findings in their manuscript fully available?

Reviewer #1: Yes

5. Is the manuscript presented in an intelligible fashion and written in standard English?

Reviewer #1: Yes

6. Review Comments to the Author

Reviewer #1: The study is interesting and didatic. All my questions have been well answered.

I have no other comments

7. PLOS authors have the option to publish the peer review history of their article (what does this mean?). If published, this will include your full peer review and any attached files.

Reviewer #1: No

---

## [Editor Report · Acceptance letter]

20 Nov 2020

PONE-D-20-16619R1 

Clinical bleeding patterns and management techniques of abnormal uterine bleeding at a teaching and referral hospital in Western Kenya. 

Dear Dr. Mutakha:

I'm pleased to inform you that your manuscript has been deemed suitable for publication in PLOS ONE. Congratulations! Your manuscript is now with our production department. 

Kind regards, 

on behalf of

Dr. Onikepe Oluwadamilola Owolabi 

Academic Editor

PLOS ONE